evolution, genomics, immunology

IgH locus, killifish, immune system, evolution, antibody

**Author for correspondence:**
Dario Riccardo Valenzano
e-mail: dvalenzano@age.mpg.de

# Extreme genomic volatility characterizes the evolution of the immunoglobulin heavy chain locus in cyprinodontiform fishes

William J. Bradshaw[1,2] and Dario Riccardo Valenzano[1,2]

[1]Max Planck Institute for Biology of Ageing, Joseph-Stelzmann-Str. 296, 50937 Cologne, Germany
[2]CECAD Research Center, University of Cologne, Joseph-Stelzmann-Str. 26, 50937 Cologne, Germany

WJB, 0000-0002-7345-3866; DRV, 0000-0002-8761-8289

The evolution of the adaptive immune system has provided vertebrates with a uniquely sophisticated immune toolkit, enabling them to mount precise immune responses against a staggeringly diverse range of antigens. Like other vertebrates, teleost fishes possess a complex and functional adaptive immune system; however, our knowledge of the complex antigen-receptor genes underlying its functionality has been restricted to a small number of experimental and agricultural species, preventing systematic investigation into how these crucial gene loci evolve. Here, we analyse the genomic structure of the immunoglobulin heavy chain (*IGH*) gene loci in the cyprinodontiforms, a diverse and important group of teleosts present in many different habitats across the world. We reconstruct the complete *IGH* loci of the turquoise killifish (*Nothobranchius furzeri*) and the southern platyfish (*Xiphophorus maculatus*) and analyse their *in vivo* gene expression, revealing the presence of species-specific splice isoforms of transmembrane *IGHM*. We further characterize the *IGH* constant regions of 10 additional cyprinodontiform species, including guppy, Amazon molly, mummichog and mangrove killifish. Phylogenetic analysis of these constant regions suggests multiple independent rounds of duplication and deletion of the teleost-specific antibody class *IGHZ* in the cyprinodontiform lineage, demonstrating the extreme volatility of *IGH* evolution. Focusing on the cyprinodontiforms as a model taxon for comparative evolutionary immunology, this work provides novel genomic resources for studying adaptive immunity and sheds light on the evolutionary history of the adaptive immune system.

The ancient evolutionary arms race between hosts and parasites has given rise to a wide variety of sophisticated immune adaptations in different taxa [1]. Among the most complex and effective of these is the vertebrate adaptive immune system, in which developing B- and T-cells generate a vast diversity of antigen-receptor sequences through dynamic recombination of their genomic sequence [1–3]. By combining this enormous diversity in antigen specificities with antigen-dependent clonal expansion and long-term immune memory [4,5], vertebrates can progressively improve their protection against recurrent immune challenges while also coping effectively with rapidly evolving pathogenic threats, dramatically improving their ability to survive and thrive in a complex immune environment.

The immunoglobulin heavy chain (*IGH*) is one of the most important antigen-receptor genes in the adaptive immune system, determining both the effector function and the majority of the antigen-specificity of the antibodies produced by each B-cell [6]. The native structure of the *IGH* gene locus has a profound effect on adaptive immunity, determining the range of gene segment

choices available for the VDJ recombination process giving rise to novel antigen-receptor sequences [2], the possible antibody classes (or *isotypes*) available, and the relationship between VDJ recombination and isotype choice [7]. Understanding the structure of this locus is essential for understanding adaptive-immune function in any given vertebrate species, while comparing loci between species can provide important insight into the adaptive immune system's complex evolutionary history.

The teleost fishes are the largest and most diverse group of vertebrates, with nearly 30 000 species comprising almost half of extant vertebrate diversity [8]. Previous work has characterized the *IGH* locus structure in a number of teleost species, including zebrafish [9], medaka [10], three-spined stickleback [11,12], rainbow trout [13], fugu [14] and Atlantic salmon [15]. These characterizations have revealed remarkable diversity in the size and structure of teleost *IGH* loci [7]. However, the number of loci characterized is very small compared to the total evolutionary diversity of teleosts, and is mainly confined to major aquaculture species and established research models. This relatively sparse sampling has prevented higher-resolution analysis of *IGH* structural evolution in teleost fishes.

Here, we present the first characterizations of *IGH* loci in the Cyprinodontiformes, a large teleost order with representatives in diverse ecological niches worldwide. Complete characterizations were performed on the loci of the turquoise killifish (*Nothobranchius furzeri*) and southern platyfish (*Xiphophorus maculatus*), two important model organisms for ecological and evolutionary research [16–19], while the loci of 10 further species (figure 1; electronic supplementary material, table S4) underwent partial characterization with a focus on their constant regions. Comparison of these loci revealed dramatic differences in *IGH* locus structure and function, including surprising differences in isotype availability and exon usage. Phylogenetic analysis suggests that the specialized mucosal isotype *IGHZ* has undergone repeated duplication and convergent loss in the course of cyprinodontiform evolution, indicating an unexpected degree of volatility in mucosal adaptive immunity. Taken together, this work significantly extends our knowledge of constant-region diversity in teleost fish, and establishes the cyprinodontiforms, and especially the African killifishes, as an ideal model system for comparative evolutionary immunology.

# 1. Results

## (a) The *IGH* loci of *N. furzeri* and *X. maculatus* are highly distinct

In order to assemble and characterize the *IGH* loci in *N. furzeri* and *X. maculatus*, published *IGH* gene segments from zebrafish [9], medaka [10] and stickleback [11,12] were aligned to the most recent genome assemblies of *N. furzeri* and *X. maculatus* (Material and methods). In *X. maculatus*, a single promising region was identified on chromosome 16, while in the *N. furzeri* genome a single region on chromosome 6 and a number of unaligned scaffold sequences were identified as potentially containing parts of the locus (electronic supplementary material, table S2). In order to determine which of the candidate scaffolds were genuine parts of the *N. furzeri IGH* locus and integrate them into a continuous

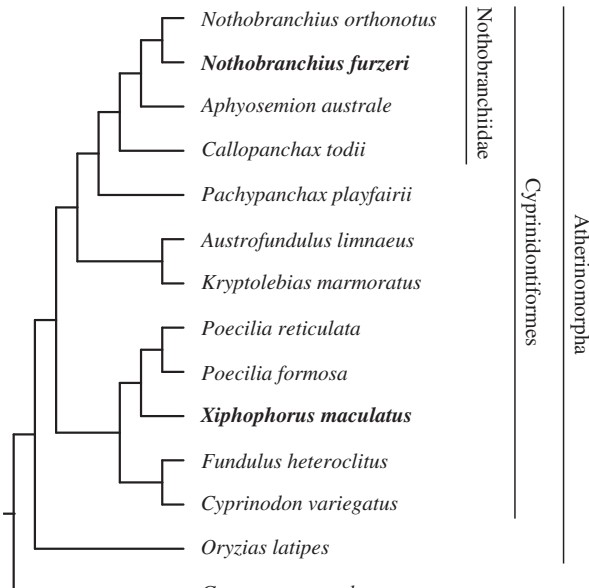

**Figure 1.** Cladogram of species included in the *IGH* locus analysis. Boldface type indicates species for which new, complete *IGH* locus assemblies were generated for this study; other species were either previously characterized reference species (*G. aculeatus*, *O. latipes*) or underwent constant-region characterization only (all other species). Labelled vertical bars designate higher taxa of interest.

locus sequence, we performed high-coverage sequencing and assembly of bacterial artificial chromosome (BAC) clones from the killifish genomic BAC library [17] whose end sequences aligned to promising genome scaffolds (electronic supplementary material, table S3). The resulting BAC inserts were integrated with the identified genome scaffolds (electronic supplementary material, figure S7) to produce a single, contiguous locus sequence, on which *IGH* gene segments were identified through more stringent alignment to sequences from reference species (electronic supplementary material, figure S7).

The *IGH* locus in *N. furzeri* occupies roughly 306 kb on chromosome 6 (NFZ v. 2.0, GenBank accession JAAVVJ010000000), while that of *X. maculatus* occupies roughly 293 kb on chromosome 16 (scaffold NC_036458.1, GenBank accession GCA_002775205.2). While similar in size, the two loci differ markedly in organization and content: while the *N. furzeri* locus comprises two distinct subloci on opposite strands (*IGH1* and *IGH2*, figure 2*a* and electronic supplementary material, figure S1), that of *X. maculatus* forms a single long configuration without any additional subloci (figure 2*b*). The two subloci of the *N. furzeri* locus exhibit a very high degree of synteny with one another in the JH and constant regions, while the VH and DH regions are more divergent (electronic supplementary material, figure S2a).

Three constant-region isotypes have been observed in previously published teleost loci: *IGHM* and *IGHD*, which are universal in teleosts and homologous to the isotypes of the same names in mammals, and *IGHZ* (also known as *IGHT*), which is teleost-specific and absent in a minority of previously published loci [7]. *X. maculatus IGH*, *N. furzeri IGH1* and *N. furzeri IGH2* all contain intact and highly similar *IGHM* and *IGHD* constant regions, with a six-exon $C_\mu1-C_\mu2-C_\mu3-C_\mu4-TM1-TM2$ configuration for *IGHM* and a 12 exon $C_\delta1-(C_\delta2-C_\delta3-C_\delta4)_2-C_\delta5-C_\delta$

Proc. R. Soc. B 287: 20200489

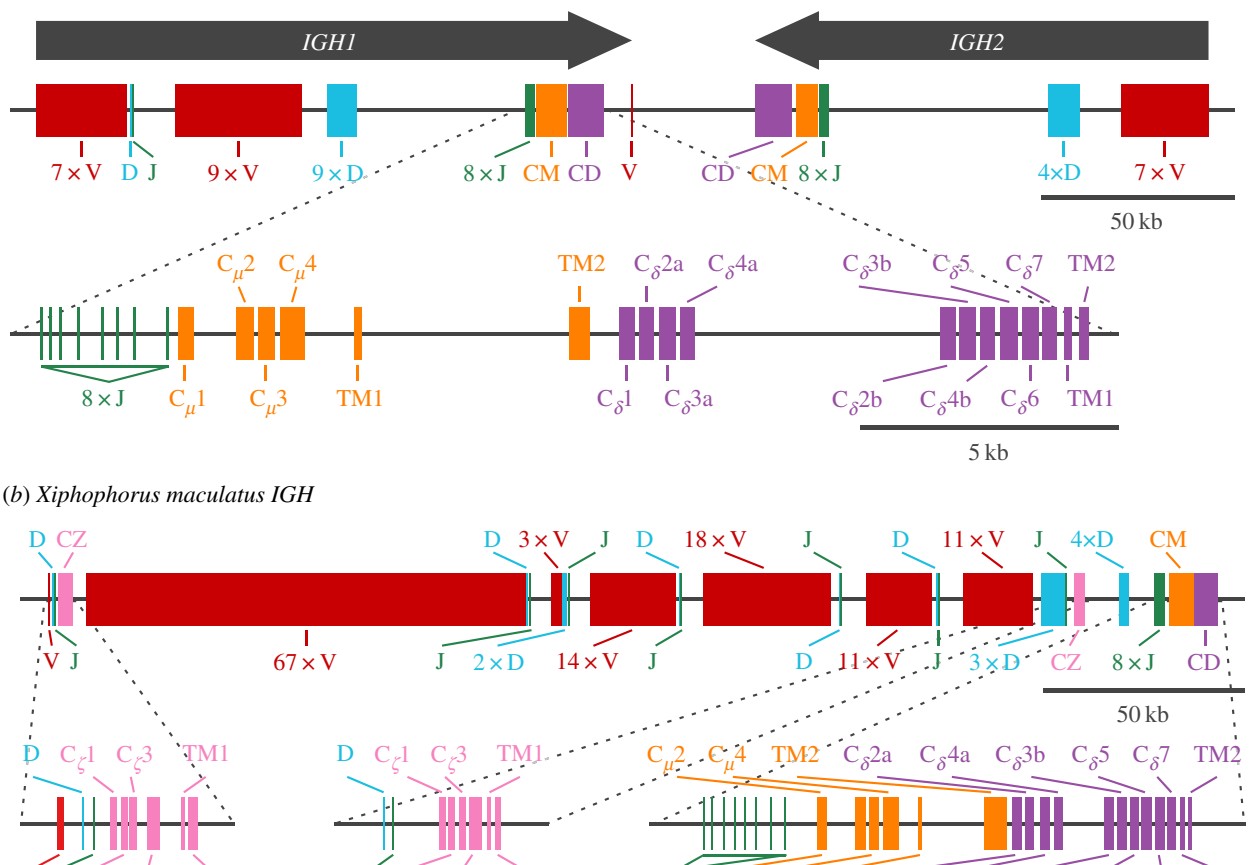

**Figure 2.** IGH locus structure in Nothobranchius furzeri and Xiphophorus maculatus. (a) Arrangement of VH, DH, JH and constant regions on the N. furzeri IGH locus, indicating the two subloci IGH1 and IGH2 and the detailed exon composition of the IGH1 constant regions. (b) Arrangement of VH, DH, JH and constant regions on the X. maculatus IGH locus, indicating the detailed exon composition of each constant region.

$6-C_\delta 7-TM1-TM2$ configuration for IGHD. Such expansion of IGHD through tandem duplications of the $C_\delta 2-C_\delta 3-C_\delta 4$ exons is common in teleost loci [7]. Secretory forms of IGHD have previously been observed in a minority of teleost loci, produced via either a specialized secretory exon [20] or a post-$C_\delta 7$ secretory tail [21]; however, neither of these configurations could be found in either N. furzeri or X. maculatus, and it may be the case that IGHD is expressed solely in transmembrane form in these species.

Previous work in rainbow trout has shown that, while IGHM is primarily responsible for the serum response to antigenic stimulus, the mucosal response is primarily mediated by IGHZ [22,23], suggesting that this isoform has a specialized mucosal role analogous to IGHA in mammals. Unlike IGHM and IGHD, however, IGHZ is completely absent from the N. furzeri IGH locus. By contrast, the X. maculatus IGH locus contains two distinct IGHZ constant regions: IGHZ1 and IGHZ2. Despite sharing a common six-exon $C_\zeta 1-C_\zeta 2-C_\zeta 3-C_\zeta 4-TM1-TM2$ configuration (figure 2b), these two paralogous constant regions are highly distinct, with an average of only 48.0% amino-acid sequence identity between corresponding $C_\zeta$ exons (electronic supplementary material, figure S2b), indicating a relatively ancient origin; in contrast, corresponding $C_\mu$ and $C_\delta$ exons in the two N. furzeri subloci exhibit an average of 100% and 98.6% sequence identity respectively (electronic supplementary

material, figure S2b), suggesting a much more recent duplication event.

In terms of the variable regions of the IGH gene, the most striking difference between the two locus assemblies is in the total number of VH segments: 125 in X. maculatus compared to only 24 in N. furzeri. In contrast, the number of DH and JH segments is similar between the two species, with 14 DH and 17 JH segments in N. furzeri and 14 DH and 15 JH in X. maculatus. In X. maculatus, only a single VH, DH and JH segment are present upstream of IGHZ1, suggesting only a single V/D/J combination is available to antibodies of this isotype; most other segments are present in six $V_n D_{1-3} J_1$ blocks between IGHZ1 and IGHZ2, with larger blocks of DH and JH segments between IGHZ2 and IGHM. This $(V-D-J)_n$-C block structure, which is also observed in N. furzeri IGH1, is in some ways intermediate between the classic translocon configuration seen in most teleost IGH loci and the multi-cluster configuration observed in sharks [24]. Note, however, that unlike the N. furzeri locus, which is assembled from a combination of a high-quality published genome and very high coverage BAC assemblies (electronic supplementary material, tables S2, S3 and S7), our X. maculatus locus assembly is based entirely on the previously published source genome; while this genome is a highly contiguous, high-quality assembly, it is possible that some regions of the variable region have been collapsed or omitted, and hence that this locus contains even more gene segments than are included here.

**Figure 3.** RNA-sequencing data reveals distinct transmembrane isoforms of *IGHM* in *X. maculatus* and *N. furzeri*. (*a*) Schematic of *IGHM* splice isoforms in different vertebrate taxa [7]. (*b,c*) Read coverage histograms and Sashimi plots of alignment and splicing behaviour of RNA-sequencing reads aligned to the *IGHM* constant regions of (*b*), *N. furzeri* and (*c*), *X. maculatus*, showing the alternative splicing of transmembrane (blue) and secreted (red) isoforms in both species and the difference in exon usage in *IGHM-TM* between species.

## (b) *N. furzeri* and *X. maculatus* express distinct forms of transmembrane *IGHM*

The six-exon genomic structure of the *IGHM* constant region is highly conserved across the jawed vertebrates, with similar configurations observed in mammals, teleosts and elasmobranchs [7]. In all these groups, the choice between secretory and transmembrane *IGHM* is made via alternative RNA splicing, with the secretory form consistently adopting a four-exon $C_\mu 1$–$C_\mu 2$–$C_\mu 3$–$C_\mu 4$ configuration. Transmembrane *IGHM*, by contrast, differs in configuration between taxa [7]: in mammals, a cryptic splice site within $C_\mu 4$ is used to connect the transmembrane exons, while in teleosts the canonical splice site at the end of $C_\mu 3$ is typically used, excising $C_\mu 4$. Unusually, the configuration of *IGHM-TM* in medaka differs from that of other teleosts, with $C_\mu 2$ spliced directly to TM1 [7,10] (figure 3*a*). Given this surprising diversity, we decided to investigate which splice isoforms are expressed in *N. furzeri* and *X. maculatus*.

To investigate the exon configuration of expressed *IGH* mRNA in *N. furzeri* and *X. maculatus*, published datasets of RNA-sequencing reads from both species (electronic supplementary material, table S5) were mapped to their respective *IGH* loci using STAR [25]. Surprisingly, the results revealed that the two species used different exon configurations for *IGHM-TM*: in *X. maculatus*, the standard teleost five-exon configuration was used (figure 3*c*), while *N. furzeri* used the unusual four-exon configuration seen in medaka (figure 3*b*; electronic supplementary material, figure S6), demonstrating that both configurations persist among the cyprinodontiforms.

In contrast to *IGHM*, both *N. furzeri* and *X. maculatus* shared a common configuration of transmembrane *IGHD*, with all 12 exons expressed in series. As in other teleosts

[7], expressed *IGHD* in both species began with a chimeric $C_\mu 1$ exon from the upstream *IGHM* constant region (electronic supplementary material, figure S4). In *X. maculatus*, meanwhile, both *IGHZ1* and *IGHZ2* expressed a six-exon transmembrane isoform, while *IGHZ1* was also found to give rise to a four-exon secreted isoform comprising $C_\zeta 1$ to $C_\zeta 4$ and a run-on secretory tail; while a tail sequence was also found following $C_\zeta 4$ in *IGHZ2*, no expression of a distinct secretory isoform was detectable in the RNA-sequencing data for this constant region (electronic supplementary material, figure S5).

## (c) *IGHZ* has undergone repeated duplication and loss in the Cyprinodontiformes

Medaka (*Oryzias latipes*) is the closest relative of either *N. furzeri* or *X. maculatus* whose *IGH* locus has previously been characterized, and one of the few teleost species previously known to lack *IGHZ* [7,10]. Despite this close relationship, the presence of multiple *IGHZ* constant regions in *X. maculatus* strongly implied that the absence of this isotype in medaka and *N. furzeri* is the result of two independent deletion events. To investigate this hypothesis, we characterized *IGH* constant-region sequences in the genomes of 10 further cyprinodontiform species (figure 1; electronic supplementary material, table S4), as well as a new and improved medaka genome assembly (GenBank accession GCA_002234675.1).

The analysed loci were highly variable, with dramatic variation in the number and arrangement of constant-region sequences (figure 4; electronic supplementary material, tables S26–S28). Of the 13 species investigated, all had at least one tandem pair of *IGHM* and *IGHD* constant regions, while eight possessed at least one complete *IGHZ*

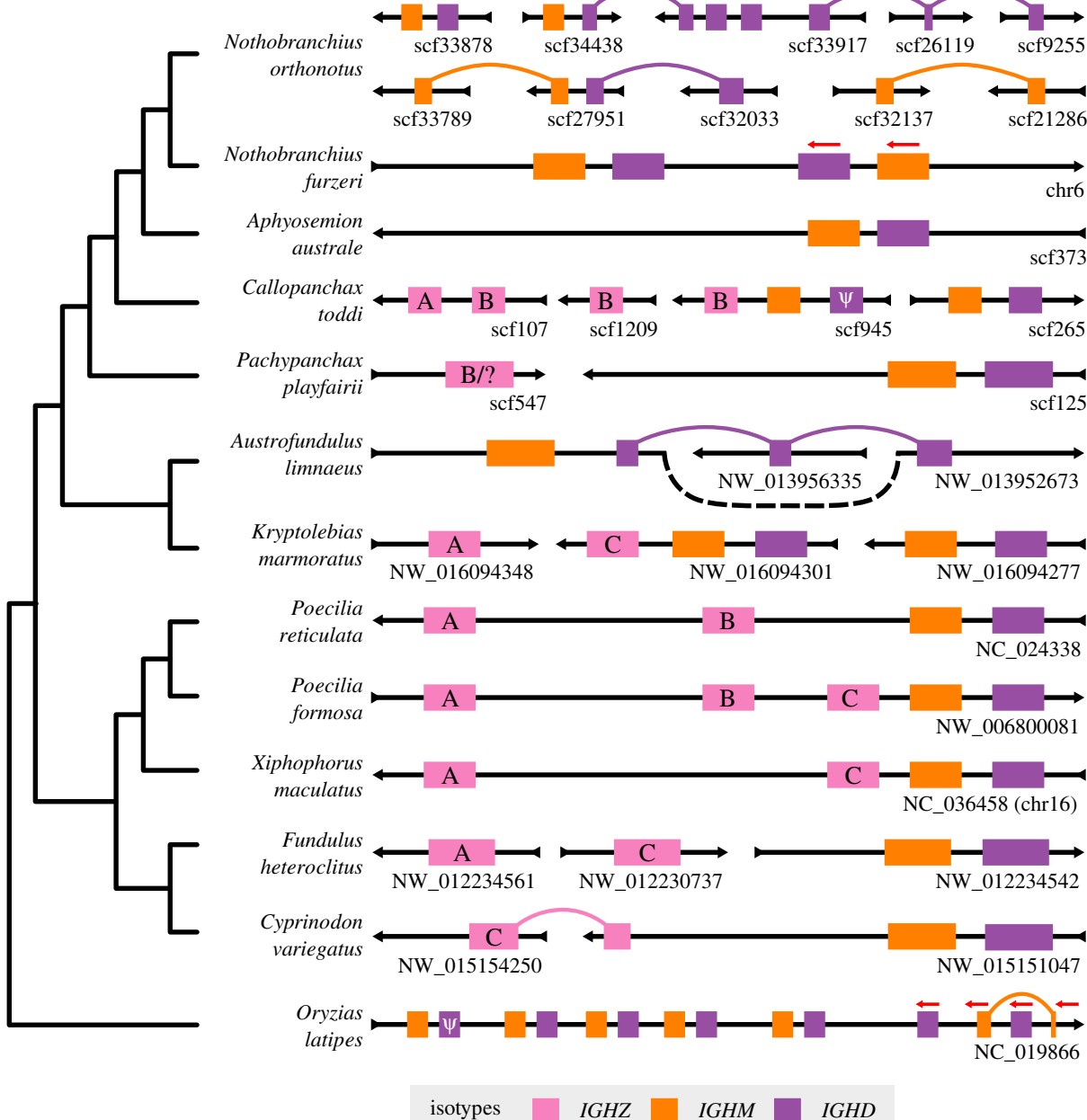

**Figure 4.** Constant-region organization in the Atherinomorpha. Schematic of newly characterized *IGH* constant regions in the genomes of 13 species from the Atherinomorpha (Cyprinodontiformes + medaka). Scaffold orientation is given by the black arrows; constant regions are oriented left-to-right unless otherwise specified (red arrows above constant regions). Scaffold names are displayed beneath each scaffold on the right-hand side. Links between regions on different scaffolds indicate that exons from what appears to be the same constant region are distributed across multiple scaffolds in the order indicated; the order of unlinked scaffolds is arbitrary. The isotype of each region is given by its colour; *IGHZ* regions are further annotated with their subclass (figure 5b). Clearly pseudogenized constant regions are indicated by $\Psi$. Isotype length, scaffold length and scaffold position are not to scale. Variable regions and lone, isolated constant-region exons are not shown. The cladogram to the left reproduces the species tree from figure 1, and is not intended to represent orthology relationships between assembled loci.

constant region (figure 4). Of the exceptions, *Austrofundulus limnaeus* was found to exhibit an orphaned, pseudogenized *IGHZ-TM1* exon but no $C_\zeta$ exons in the current genome assembly, while no *IGHZ* exons at all were found in the genomes of *O. latipes*, *N. furzeri*, *Aphyosemion australe*, or *Nothobranchius orthonotus*. Assuming that *IGHZ*, once deleted, cannot be restored to the *IGH* locus in a lineage, a simple visualization on a species tree (figure 5a) confirms that medaka and *N. furzeri* represent two distinct *IGHZ* deletion events; *A. limnaeus* appears to represent another independent deletion event, for a total of at least three *IGHZ* deletions within the clade containing the cyprinodontiforms and medaka.

In addition to being lost repeatedly, *IGHZ* also demonstrated a high level of multiplicity within the cyprinodontiforms, with a geometric mean of 1.93 *IGHZ* constant regions per *IGHZ*-bearing locus, suggesting a more complex history than can be captured by a simple presence/absence metric. Concordantly, phylogenetic analysis with PRANK [26] and RAxML [27] (figure 5b, alignment length 1733 bp, 35% gaps/missing characters) indicates three distinct monophyletic clades (or subclasses) of *IGHZ* constant regions (*IGHZA* to *C*), each of which is present in multiple species and appears to have been present in the common ancestor of the eight IGHZ-bearing species analysed. The only locus whose *IGHZ* could not be assigned to one of these

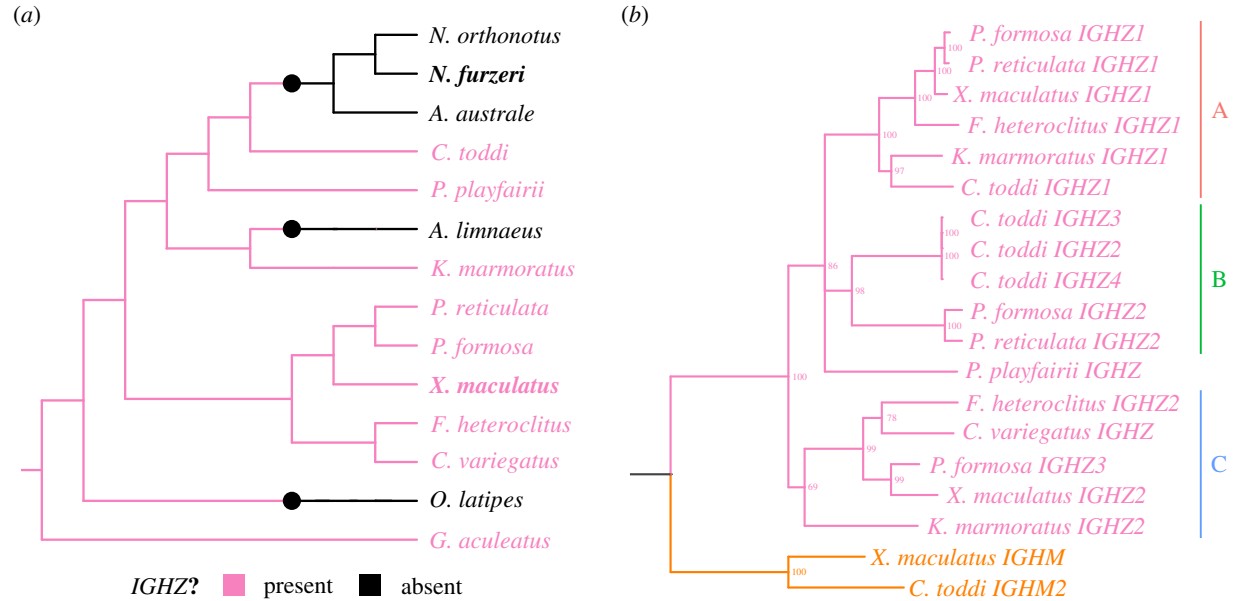

**Figure 5.** *IGHZ* has undergone repeated duplication and loss in the Cyprinodontiformes. (*a*) Cladogram of species from figure 1, with three-spined stickleback (*Gasterosteus aculeatus*) as the outgroup, coloured according to known *IGHZ* status. Large coloured points indicate inferred state-change events. (*b*) Phylogram of concatenated $C_\zeta 1$–4 nucleotide sequences from *n IGHZ*-bearing cyprinodontiform species, with $C_\mu 1$–4 sequences from two species as outgroup (in orange at the bottom). Nodes with less than 65% bootstrap support are collapsed into polytomies, while major monophyletic subclasses are annotated on the right.

subclasses, that of *Pachypanchax playfairii*, appears to have undergone a fusion event, with *P. playfairii* $C_\zeta 1$ and $C_\zeta 2$ aligning to *IGHZB* exons from other species while *P. playfairii* $C_\zeta 3$ and $C_\zeta 4$ show more ambiguous alignment behaviour favouring *IGHZA* or *IGHZC* (electronic supplementary material, figure S3).

In summary, in addition to *IGHM* and *IGHD*, the cyprinodontiforms appear to have ancestrally possessed at least three subclasses of *IGHZ*, which subsequently evolved in parallel across the clade. Each of these subclasses has been lost in multiple species, with different species showing distinct patterns of retention and loss, and in at least one lineage—that of *P. playfairii*—two different *IGHZ* lineages appear to have fused to produce a chimeric constant region. All three subclasses are missing from a subset of species in the Nothobranchiidae (including *N. furzeri*), and also appear to have been independently lost in *Austrofundulus limnaeus*, demonstrating the remarkable volatility of the *IGH* locus across evolutionary time.

## 2. Discussion

The immunoglobulin heavy chain locus is notable for its size and complexity, as well as for the central role it plays in vertebrate immunity and survival. Previous research in teleost fishes has revealed a remarkable degree of diversity in the length and organization of different *IGH* loci [7], with important but understudied implications for antibody diversity and immune functionality among teleost species.

In this study, we presented the first detailed characterizations of *IGH* loci from the Cyprinodontiformes, a widespread order of teleost fishes that include many important model systems in evolutionary biology and ecology. Two such loci, those of the turquoise killifish *N. furzeri* and the southern platyfish *X. maculatus*, underwent complete assembly and characterization, while 10 other species received partial characterizations. These additional species were selected on the basis

of their relatedness to *N. furzeri* and *X. maculatus* and their prevalence in the literature, and included a number of prominent ecological model organisms (including guppy [28], mummichog [29] and mangrove rivulus [30]), yielding a dataset with significant relevance to researchers studying the role of infection and immunity in teleost ecology.

The *IGH* loci of *X. maculatus* and *N. furzeri* exhibited radically different organizations, with dramatic differences in VDJ configuration and isotype availability. These results are consistent with previous findings in teleost loci and support a process of rapid evolution in antigen-receptor genes. Characterization of the constant regions of additional species confirmed this finding, with several groups of closely related species (e.g. *N. furzeri*, *N. orthonotus* and *Callopanchax toddi*) showing highly divergent locus structures (figure 4).

It is interesting to speculate on the origins of this extremely rapid diversification. Very little is known about the relationship between environmental context and immune locus structure; it is possible that part of the variety in *IGH* gene locus structure in the Cyprinodontiformes represents divergent adaptations to different immune environments. Alternatively, this diversification may be primarily the result of unusually high rates of stochastic, non-adaptive changes in *IGH* gene structure, or to relaxation of selective constraints. Finally, at least some of the difference between locus structures in different species is likely to be attributable to differences in assembly quality; for example, the characterization of medaka constant regions presented here contains many fewer unusual or incomplete constant regions than that presented in the published medaka *IGH* locus [10], primarily due to the increased quality of more recent genome assemblies. Issues with assembly quality could also account for the apparent complexity of the *N. orthonotus* locus, as the genome of this species was assembled from a wild-caught individual with a high degree of heterozygosity [31].

The teleost-specific isotype *IGHZ* is widespread among teleost species, and appears to play a specialized role in mucosal immunity [22,23]. Before the publication of this

work, only two teleost species (medaka and channel catfish) were known or thought to lack *IGHZ*, suggesting that the loss of this isotype may be a relatively rare event. However, in addition to confirming the absence of *IGHZ* in medaka, the work presented here has identified four new teleost species (*N. furzeri*, *N. orthonotus*, *Aphyosemion australe* and *Austrofundulus limnaeus*) that appear to lack *IGHZ* constant regions, representing two previously unknown loss events independent from that affecting the closely related medaka. This finding, which triples the number of known teleost species without *IGHZ* and doubles the number of known loss events, is even more striking when combined with the discovery that the cyprinidontiform common ancestor likely had no fewer than three distinct *IGHZ* constant regions (figure 5*b*), all of which would have to be lost on the way to any *IGHZ*-free lineage. Taken together, these observations suggest that the presence/absence of *IGHZ* in teleost *IGH* loci may be much more volatile than was previously apparent, and raises the possibility that, given sufficiently high-density analysis, a surprisingly high frequency of *IGHZ*-lacking species may also be found in other teleost lineages.

The absence of *IGHZ* from so many species naturally raises the question of how mucosal adaptive immunity in these species differs from that of their *IGHZ*-bearing relatives. How, and to what extent, can *IGHM* (or perhaps even *IGHD*) compensate for the loss of this specialized mucosal antibody class? Is such compensation even necessary, or do these species rely on some other form of mucosal immunity? These questions are especially interesting in the case of *IGHZ*-lacking species with close *IGHZ*-bearing relatives (e.g. *N. furzeri* and *Callopanchax toddi*, or *Austrofundulus limnaeus* and *Kryptolebias marmoratus*); if it is the case that mucosal immune responses differ systematically between these species, such that other isotypes take up some or all of the roles normally played by *IGHZ*, then uncovering the mechanisms by which this shift is regulated could reveal important new insights into decision-making and control of humoral adaptive immunity. Similarly, characterizing the different functional roles and responses of different *IGHZ* subclasses in cyprinodontiform fishes could yield important information about how these species interact with different aspects of their immune environment.

One of the most important advances in immunology in recent years has been the explosion of quantitative, high-throughput approaches for investigating the composition, diversity and functionality of the antibody repertoire [32,33]. As a direct result of the research presented here, 12 previously uncharacterized teleost species now have databases of *IGH* constant-region sequences available, enabling these immuno-globulin-sequencing approaches to be applied in the cyprinodontiforms for the first time. Combining antibody-repertoire data with other information gathered from wild fishes could yield important new insights into the role of the adaptive immune system in the lives and evolution of wild vertebrates, as well as providing an opportunity to independently validate the locus assemblies presented here. In addition, the possibility of sequencing the repertoires of several related species adds an exciting comparative dimension previously missing in immune-repertoire studies, opening up the possibility of simultaneously comparing the response of different closely related species to a common immunogenic stimulus. This comparative element would be especially interesting when investigating the repertoire responses of closely related species with different *IGHZ* genotypes, as well as when comparing the functional roles of different *IGHZ* subclasses across species. Such large-scale comparative repertoire studies provide a novel opportunity for comparative evolutionary immunology in the Cyprinodontiformes.

The *IGH* locus is not the only immune gene notable for its size, complexity, and diversity across teleost species. Previous work on teleost MHC loci [34–36] has also revealed a great deal of complexity, including a complex pattern of gene-lineage diversification and loss. Future studies considering both these genes, perhaps alongside T-cell receptor loci [36,37] could yield more general insights into the effects of immune gene evolution on immune functionality in teleosts. Such an approach, for which our findings here lay vital groundwork, has the potential to greatly expand our understanding of the interaction between ecological conditions and the evolution of the vertebrate adaptive immune system.

## 3. Material and methods

Comprehensive bioinformatic methods for this study can be found in the supplementary material.

### (a) Assembling the *N. furzeri IGH* locus

To identify candidate locus sequences, VH, JH and CH sequences from three reference *IGH* loci (zebrafish [9], medaka [10] and three-spined stickleback [11,12]) were aligned to the most recent assembly of the *N. furzeri* genome [38] (NFZ v. 2.0, GenBank accession JAAVVJ010000000) using BLAST [39]. Scaffolds that did not align to any *IGH* gene segments were discarded, while those containing promising alignments to at least two distinct types of gene segment were retained as candidates. Scaffolds containing promising alignments to only one type of gene segment were only retained if these alignments covered at least 1% of the total length of the scaffold; this weak cut-off was intended to eliminate long scaffolds and superscaffolds containing a small number of orphaned gene segments.

In order to determine which candidate scaffolds contained parts of the *IGH* locus and integrate them into a single sequence, clones from the killifish genomic BAC library [17] were identified on the basis of alignment of their end sequences to promising genome scaffolds. These BAC clones were provided to us by the Leibniz Institute on Aging in Jena, Germany and isolated and sequenced as described in the next section.

Following sequencing, demultiplexed and adapter-trimmed MiSeq reads were processed with Trimmotatic [40] to trim low quality sequence and Bowtie 2 [41] to remove contaminating *E. coli* sequences, then corrected with QuorUM [42] or BayesHammer [43,44] and assembled with SPAdes [43]. Following assembly, any *E. coli* scaffolds resulting from residual contaminating reads were identified by aligning scaffolds to the *E. coli* genome using BLASTN [39], and scaffolds containing significant matches were discarded. The remaining scaffolds were then scaffolded using SSPACE [45] using jumping libraries from the killifish genome project [16,17,38].

The two scaffolded assemblies (using BayesHammer- and QuorUM-corrected reads, respectively) agreed on the order and orientation of contigs at many points, but disagreed at others. In order to guarantee the reliability of the assembled scaffolds, the assemblies were broken into segments, such that the contiguity of each segment was agreed on between both assemblies. To integrate these segments into a contiguous insert assembly, points of agreement between BAC assemblies from the same genomic region (e.g. two scaffolds from one assembly aligning concordantly to one scaffold from another) and between BAC assemblies and genome scaffolds, were used to combine scaffolds where possible. Any

still-unconnected scaffolds were assembled together through pair-wise end-to-end PCR using Kapa HiFi HotStart ReadyMix PCR Kit according to the manufacturer's instructions, followed by Sanger sequencing [46] (Eurofins). PCR primers for end-to-end PCR were designed using Primer3 [47].

Following BAC insert assembly, assembled inserts were screened for *IGH* locus segments in the same manner described for genome scaffolds above. Passing BAC inserts were aligned to the candidate genome scaffolds and chromosome sequence with BLASTN and integrated manually (electronic supplementary material, figure S7), giving priority in the event of a sequence conflict to (i) any sequence containing a gene segment missing from the other, and (ii) the genome scaffold sequence if neither sequence contained such a segment. The orientation of BACs and unplaced genome scaffolds integrated into the assembly was selected on the basis of their alignment to the main locus sequence on chromosome 6, or to other sequences that had already been so integrated. BACs and scaffolds which could not be integrated into the locus sequence in this way were discarded.

## (b) BAC isolation and sequencing

All BAC clones that were sequenced for this research were provided by the Leibniz Institute on Aging in Jena, Germany as plate or stab cultures of transformed *E. coli*, which were replated and stored at 4°C. Prior to isolation, the clones of interest were cultured overnight in at least 100 ml LB medium. The resulting liquid cultures were transferred to 50 ml conical tubes and centrifuged (10–25 min, 4°C, 3500*g*) to pellet the cells. The supernatant was carefully discarded and the cells were resuspended in 18 ml QIAGEN buffer P1.

After resuspension, the cultures underwent alkaline lysis to release the BAC DNA and precipitate genomic DNA and cellular debris. Eighteen millilitres QIAGEN buffer P2 was added to each tube, which was then mixed gently but thoroughly by inversion and incubated at room temperature for 5 min. Eighteen millilitres ice-chilled QIAGEN neutralization buffer P3 was added to precipitate genomic DNA and cellular debris, and each tube was mixed gently but thoroughly by inversion and incubated on ice for 15 min. The tubes were then centrifuged (20–30 min, 4°C, 12 000*g*) to pellet cellular debris and the supernatant was transferred to new conical tubes. This process was repeated at least two more times, until no more debris was visible in any tube; this repeated pelleting was necessary to minimize contamination in each sample, as the normal column- or paper-based filtering steps used during alkaline lysis resulted in the loss of the BAC DNA.

Following lysis, the DNA in each sample underwent isopropanol precipitation: 0.6 volumes of room-temperature isopropanol were added to the clean supernatant in each tube, followed by 0.1 volumes of 3 mol sodium acetate solution. Each tube was mixed well by inversion, incubated for 10–15 min at room temperature, then centrifuged (30 min, 4°C, 12 000*g*) to pellet the DNA. The supernatant was discarded and the resulting DNA smear was 'resuspended' in 1 ml 100% ethanol and transferred to a 1.5 ml tube, which was re-centrifuged (5 min, 4°C, top speed) to obtain a concentrated pellet. Finally, the pelleted samples were resuspended in QIAGEN buffer EB and purified of proteins and RNA using standard phenol–chloroform extraction and ethanol precipitation techniques.

The resuspended BAC isolates were sent to the Cologne Center for Genomics, where they underwent Illumina Nextera XT library preparation and were sequenced on an Illumina MiSeq sequencing machine (MiSeq Reagent Kit v. 3, 2 × 300 bp reads).

## (c) Identifying locus scaffolds in other species

Candidate *IGH* locus sequences in other species (electronic supplementary material, table S4) were identified in the same manner as for *N. furzeri*, by aligning VH, JH and CH sequences

from reference species to available genome sequences with BLAST. In the case of *X. maculatus*, the reference species used were zebrafish, stickleback, medaka and *N. furzeri*, while for all other species the gene segments from the *X. maculatus* locus were also used. Additional sequence refinement with BAC inserts was not necessary in these species.

## (d) Characterizing constant-region sequences and expression

Constant-region sequences were identified by mapping CH sequences from reference species to candidate sequences using BLAST. Overlapping alignments to reference segments of the same isotype and exon number were collapsed together, and alignment groups with a very poor maximum E-value (>0.001) were discarded, as were alignment groups overlapping with a much better alignment to a different isotype or exon type (with a bitscore difference of at least 16.5). Surviving alignment groups then underwent a second step of at increased stringency, requiring an E-value of at most $10^{-8}$ and at least two aligned reference exons over all reference species to be retained. Exon sequences corresponding to surviving alignment groups were extracted into FASTA format and underwent manual curation to resolve conflicts, assign exon names and refine intron/exon boundaries (electronic supplementary material, tables S6 and S15). In order to validate intron/exon boundaries and investigate splicing behaviour among *IGH* constant-region exons in *N. furzeri* and *X. maculatus*, published RNA-sequencing data comprising short unassembled Illumina reads (electronic supplementary material, table S5) were aligned to the annotated locus using STAR [25]. In both cases, reads files from multiple individuals were concatenated and aligned together, and the *IGH* locus was masked using RepeatMasker Open-4.0 prior to mapping. Mapped reads spanning predicted exons of more than 10 kb were excluded from the alignment, as were read pairs mapping more than 10 kb apart. Following alignment, the resulting SAM files were processed into sorted, indexed BAM files using SAMtools [48] and visualized with Integrated Genomics Viewer [49] to determine intron/exon boundaries and major splice isoforms in each dataset. Read-coverage and Sashimi plots (figure 3; electronic supplementary material, figure S4–S6) were generated using Gviz [50].

For species other than *N. furzeri* or *X. maculatus*, intron/exon boundaries were predicted manually based on BLASTN and BLASTP alignments to closely related species and the presence of conserved splice-site motifs [51]. In cases where no 3′ splice site was expected to be present, the nucleotide exon sequence was terminated at the first canonical polyadenylation site [52], while the amino-acid sequence was terminated at the first stop codon. In many cases, it was not possible to locate a TM2 exon due to its very short conserved coding sequence [9,11].

## (e) Characterizing variable-region sequences

For VH and JH segments, segments from reference species were used to construct a multiple-sequence alignment with PRANK [26], which was then used by NHMMER [53] to perform a Hidden–Markov Model-based search for matching sequences in the locus. Sequence candidates were extended on either end to account for boundary errors, then refined manually. In the case of VH sequences, 3′ ends were identified by the start of the RSS heptamer sequence [54], if present, while 5′ ends, FR/CDR boundaries were identified using IMGT/DomainGapAlign [55,56]. For JH segments, 5′ ends were identified using the RSS heptamer sequence, while the 3′ end was identified using the conserved splice-junction motif GTA.

Following manual curation, VH segments were grouped into families based on their pairwise sequence identity, as determined

by Needleman–Wunsch global alignment [57] and single-linkage hierarchical clustering, with a clustering threshold at 80% sequence identity (electronic supplementary material, figures S8–S10). VH segments were named based on family membership and position in the locus; JH segments were named based on position only (electronic supplementary material, tables S13 and S24). DH segments were located using their distinctive pattern of flanking recombination signal sequences in opposite sense [3]. Potential matches to this pattern were searched for using EMBOSS FUZZNUC [58], with a high mismatch tolerance (up to eight mismatches across the whole pattern). Promising candidate sequences from this search were oriented based on the orientation of flanking VH or JH sequences on the same scaffold, then underwent a second, more stringent filtering step in which sequences lacking the most conserved positions in each RSS [54] were discarded. Surviving candidates were checked and curated manually, and those passing manual curation were named based on their position in the locus (electronic supplementary material, tables S10 and S22).

## (f) Phylogenetic inference

Cladograms of teleost species (figures 1 and 5a) were constructed using information from Cui *et al.* [31] and Hughes *et al.* [59] and visualized using ggtree [60].

To construct a phylogram of *IGHZ* sequences (figure 5b), the nucleotide sequences of $C_\zeta$1-4 exons from each *IGHZ* constant region were concatenated together, and the different constant regions were aligned using PRANK [26]. The resulting multiple-sequence alignment was then used to perform phylogenetic inference with RAxML [27], using the SSE3-enabled parallelized version of the software, the standard GTR-Gamma nucleotide substitution model, and built-in rapid bootstrapping with 1000 bootstrap replicates; during tree inference, the third codon position was partitioned into a separate model. The bootstrap-annotated `RAxML_bipartitions` file was inspected and rooted manually in Figtree (v. 1.4) and again visualized using ggtree; during tree visualization, nodes with bootstrap support of less than 65% were collapsed into polytomies.

## (g) Inter- and intralocus sequence comparison

Synteny between subloci in the *N. furzeri* locus (electronic supplementary material, figure S2a) was analysed using the standard synteny pipeline from the DECIPHER R package [61], which searches for chains of exact *k*-mer matches within two sequences.

Comparison between constant-region exons, either within the same locus (electronic supplementary material, figure S2b) or between loci (electronic supplementary material, figure S3) were performed using Needleman–Wunsch exhaustive global alignments [57], as implemented in the Biostrings R package, using the default scoring parameters from that package.

Data accessibility. All data are accessible and the links are provided in the manuscript.

Authors' contributions. W.J.B. and D.R.V. conceived the study and designed the research. W.B. performed all the analyses. Both authors wrote the manuscript.

Competing interests. We declare we have no competing interest.

Funding. This work was funded by the Max Planck Institute for Biology of Ageing, the Cologne Graduate School of Ageing Research, the Max Planck Society and the DFG Collaborative Research Center 1310.

Acknowledgements. We thank Kathrin Reichwald for providing the BAC clones used in this study; Mario Ventura and Nicola Lorusso for early help and support with BAC isolation; Bérénice Benayoun, Anton Korobeynikov, Jorge Boucas, Franziska Metge and Bernd Wozny for help and advice with the BAC sequence assembly process; and David Willemsen and Rongfeng Cui for critically reading and reviewing the manuscript.

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
