## [Reviewer comments · Proceedings of the Royal Society B: Biological Sciences]

Review History

RSPB-2019-2180.R0 (Original submission)

Review form: Reviewer 1 (Brian Dixon)

Recommendation

Accept with minor revision (please list in comments)

Scientific importance: Is the manuscript an original and important contribution to its field?

Excellent

General interest: Is the paper of sufficient general interest?

Good

Quality of the paper: Is the overall quality of the paper suitable?

Excellent

Is the length of the paper justified?

Yes

Should the paper be seen by a specialist statistical reviewer?

No

Do you have any concerns about statistical analyses in this paper? If so, please specify them explicitly in your report.

No

It is a condition of publication that authors make their supporting data, code and materials available - either as supplementary material or hosted in an external repository. Please rate, if applicable, the supporting data on the following criteria.

Is it accessible?

Yes

Is it clear?

Yes

Is it adequate?

Yes

Do you have any ethical concerns with this paper?

No

Comments to the Author

This is a fascinating and well conducted study of the variation in genome content of IgM genes in several species of cyprinodontiformes. The authors careful analysis shows variation in structure and expression of TM and secreted forms within single species and between clades.

This not the only study of genome content variation in immune genes of fish – the authors should see the work of these authors on MHC and other immune gene families in zebrafish: McConnell and de Jong of Chicago, Jeff Yoder of North Carolina. It would be fascinating to see if there are parallels in the stories of gene loss and the effects on subsequent immune function in general.

Small notes:

On page 2 in the last paragraph, the authors say both species IgH loci are on chromosome 16. Isn't *N. furzeri* on chromosome 6?

On page 3 in paragraph 2 there is a problem on line 13 “fillatreau2013astounding” makes no sense.

On page 5 and 7 the authors refer to the IgM locus as “primitive”, but it is clearly highly evolved, complex and flexible, so I am not sure it deserves that designation.

Review form: Reviewer 2

Recommendation

Reject – article is scientifically unsound

Scientific importance: Is the manuscript an original and important contribution to its field?

Acceptable

General interest: Is the paper of sufficient general interest?

Good

Quality of the paper: Is the overall quality of the paper suitable?

Marginal

Is the length of the paper justified?

Yes

Should the paper be seen by a specialist statistical reviewer?

No

Do you have any concerns about statistical analyses in this paper? If so, please specify them explicitly in your report.

Yes

It is a condition of publication that authors make their supporting data, code and materials available - either as supplementary material or hosted in an external repository. Please rate, if applicable, the supporting data on the following criteria.

Is it accessible?

No

Is it clear?

No

Is it adequate?

No

Do you have any ethical concerns with this paper?

No

Comments to the Author

The authors present detailed information on the structure of the immunoglobulin heavy chain (IGH) locus in two cyprinodontiform fishes, and a less detailed analysis of the constant region in a broader suite of fishes of this family. The main focus area of the study is an important one for understanding adaptive immunity in vertebrates. While I applaud the authors on the work they have done to characterize IGH loci, I have several major concerns with the analyses and their interpretation outlined below.

Major comments:

(1) The description of bioinformatic approaches for determining orthology is problematic and incomplete. Which BLAST algorithm was used? What were threshold values for initial searches? BLAST is notoriously bad for determining orthology, particularly when low complexity regions are involved. A much better approach would be using broader-scale syntenic information and proper orthology searches (e.g., OrthoMCL, Orthofinder, etc.). For example, are formal analysis of synteny with sequenced genomes would be very useful. Additionally, are there copy number variants / tandem duplications of IGH loci?

(2) The alternative splicing analysis is problematic. Short Illumina fragments require assembly and are notoriously prone to errors, particularly in repetitive regions. A much better approach is to use long read, single molecule DNA sequencing technology (nanopore sequencing or minimally), such that sequence assembly can be simplified or altogether avoided.

(3) The genomes used for the analysis are not available and thus it is not possible to review the methods used here. How complete are these assemblies? What technology was used for sequencing? How complete were the gene models (e.g., BUSCO percentage?).

Minor Comments:

The title is misleading. The data are based solely on cyprinodontiform fishes, rather than a broad survey of IGH across teleost fishes. While the current title and abstract as written will likely generate a broader readership, neither are accurate with respect to the true focus of the paper.

Constant regions should be precisely defined.

Review form: Reviewer 3

Recommendation

Accept with minor revision (please list in comments)

Scientific importance: Is the manuscript an original and important contribution to its field?

Good

General interest: Is the paper of sufficient general interest?

Good

Quality of the paper: Is the overall quality of the paper suitable?

Excellent

Is the length of the paper justified?

Yes

Should the paper be seen by a specialist statistical reviewer?

No

Do you have any concerns about statistical analyses in this paper? If so, please specify them explicitly in your report.

No

It is a condition of publication that authors make their supporting data, code and materials available - either as supplementary material or hosted in an external repository. Please rate, if applicable, the supporting data on the following criteria.

Is it accessible?

No

Is it clear?

Yes

Is it adequate?

Yes

Do you have any ethical concerns with this paper?

No

Comments to the Author

Dear authors,

Your manuscript presents well and describes a very interesting story. I have no major concerns regarding your study and my comments are minor.

Summarized they are:

Please address the issue of assembly errors in more detail in results/methods in addition to the part in the discussion. The IgH locus (loci) is tricky to handle for assembly algorithms and the V regions especially are vulnerable to collapse/breakage.

Please add detailed parameters for all bioinformatical approaches either in the main method or in the supplement.

Please consider your color scheme – it is somewhat strong on the eye and certain figures would be unreadable for colorblind persons.

Main text

Figure 2: this figure, although informative, is a bit hard to read. Color scheme is one element. The other are the expanded views. I would also like a detailed view of constant D and M on IgH2 in *N. furzeri*.

The reverse IgH2 region in *N. furzeri* could be interpreted as an assembly fault considering the data from the other closely related species and the sequence overview figure in the supplement. Could you add a sentence or two in the methods/results on how you proceeded to choose the most likely orientation of clones etc. in that region. This should also be reflected in the results/discussion as misassembly would affect some of the minor conclusions in your manuscript.

“Such expansion of IGHD through tandem duplications of the Cd2Cd3Cd4 exons is common in teleost loci fillatreau2013anstonishing” - Reference fillatreau2013anstonishing? Error in reference formatting?

Do you think that the number of V regions in *N. furzeri* is linked to the alternative assembly strategy? Have you by chance filtered out V regions in some way?

Consider toning down the conclusion on the three distinct lineages of IgHZ as there is a fork in your tree right at the base of two lineages: “suggests three distinct monophyletic clades of IgHZ”.

Figure 6 may be not be the most intuitive to some. Maybe put it in supplement or have the alignment in the supplement so the reader can visualize?

When discussing the loss of IgHZ and mucosal immunity, what about IgD – if IgHs are so variable in their evolution, could not IgD maybe obtain a role related to mucosal linings? Also considering the genomic plasticity of fish immune systems gradually revealed through the last decade or so, could fish perhaps do without mucosal immunity as we know it? Recently, I saw a manuscript on BioRxiv by Mirete-Bachiller et al. presenting fish without IgH at all – how would they handle mucosal immunity? <https://doi.org/10.1101/793695>

Supplement:

Please add a “list of content” to the supplementary file for easier navigation.

I would like you to replicate figure S2 to for the IgH2 region as well as for IgM. I realize that the 100 % identity of some regions will make it difficult, but this region could be marked in the figure maybe?

Methods:

Please update accession where applicable - NFZ v2.0, Accession TBD

The majority of the bioinformatical methods described are lacking a description of parameters used etc. Either add to the manuscript or write the details in the supplement.

What is the underlying argument for retaining hits covering 1 % of the scaffold? In this case, you are eliminating all shorter scaffold. Your main focus is on constant regions and these tend to assemble well. However, you will likely lose some variable regions using this cutoff.

Please write out the full name of FLI in Jena.

A bit unclear: “In order to guarantee the reliability of the assembled scaffolds, the assemblies produced with BayesHammer- and Quorum-corrected reads were compared, and scaffolds were broken into segments whose contiguity was agreed on between both assemblies”. Please rephrase. I assume they were broken based on contiguous sections, but not quite sure.

When using the BAC clones to adjust your assemblies, did the BAC genomic structure take precedence over the genome assembly structure when verifying sequence within a scaffold? Or did you just use this strategy to combine scaffolds.

How many scaffolds were you not able to join in the end?

Decision letter (RSPB-2019-2180.R0)

24-Dec-2019

Dear Dr Valenzano:

I am writing to inform you that your manuscript RSPB-2019-2180 entitled "Extreme genomic volatility characterises the evolution of the immunoglobulin heavy chain locus in teleost fishes" has, in its current form, been rejected for publication in Proceedings B.

This action has been taken on the advice of referees, who have recommended that substantial revisions are necessary. In particular, one reviewer expressed significant concerns with the BLAST and alternative splicing analyses that must be addressed prior to further consideration. With this in mind we would be happy to consider a resubmission, provided the comments of the referees are fully addressed. However please note that this is not a provisional acceptance.

Sincerely,
Dr Sarah Brosnan
<mailto:proceedingsb@royalsociety.org>

Associate Editor
Board Member: 1
Comments to Author:

Your manuscript was reviewed by three experts and myself. While we all found the topic of interest, as you will see, there is some disagreement among the reviewers regarding the soundness of the analyses presented and their interpretation.

Two reviewers noted that the data were not accessible, including accession. This needs to be added should the paper be resubmitted.

Please pay careful attention to the concerns raised regarding the use of BLAST and the alternative splicing analysis. Again, multiple reviewers commented on the need for additional information

on the bioinformatics approaches used, and one reviewer suggested that the authors could do a better job of placing the work in context of other studies on immune genes in fishes.

Reviewer(s)' Comments to Author:

Referee: 1

Comments to the Author(s)

This is a fascinating and well conducted study of the variation in genome content of IgM genes in several species of cyprinodontiformes. The authors careful analysis shows variation in structure and expression of TM and secreted forms within single species and between clades.

This not the only study of genome content variation in immune genes of fish – the authors should see the work of these authors on MHC and other immune gene families in zebrafish: McConnell and de Jong of Chicago, Jeff Yoder of North Carolina. It would be fascinating to see if there are parallels in the stories of gene loss and the effects on subsequent immune function in general.

Small notes:

On page 2 in the last paragraph, the authors say both species IgH loci are on chromosome 16. Isn't *N. furzeri* on chromosome 6?

On page 3 in paragraph 2 there is a problem on line 13 “fillatreau2013astounding” makes no sense.

On page 5 and 7 the authors refer to the IgM locus as “primitive”, but it is clearly highly evolved, complex and flexible, so I am not sure it deserves that designation.

Referee: 2

Comments to the Author(s)

The authors present detailed information on the structure of the immunoglobulin heavy chain (IGH) locus in two cyprinodontiform fishes, and a less detailed analysis of the constant region in a broader suite of fishes of this family. The main focus area of the study is an important one for understanding adaptive immunity in vertebrates. While I applaud the authors on the work they have done to characterize IGH loci, I have several major concerns with the analyses and their interpretation outlined below.

Major comments:

(1) The description of bioinformatic approaches for determining orthology is problematic and incomplete. Which BLAST algorithm was used? What were threshold values for initial searches? BLAST is notoriously bad for determining orthology, particularly when low complexity regions are involved. A much better approach would be using broader-scale syntenic information and proper orthology searches (e.g., OrthoMCL, Orthofinder, etc.). For example, are formal analysis of synteny with sequenced genomes would be very useful. Additionally, are there copy number variants / tandem duplications of IGH loci?

(2) The alternative splicing analysis is problematic. Short Illumina fragments require assembly and are notoriously prone to errors, particularly in repetitive regions. A much better approach is to use long read, single molecule DNA sequencing technology (nanopore sequencing or minimally), such that sequence assembly can be simplified or altogether avoided.

(3) The genomes used for the analysis are not available and thus it is not possible to review the

methods used here. How complete are these assemblies? What technology was used for sequencing? How complete were the gene models (e.g., BUSCO percentage?).

Minor Comments:

The title is misleading. The data are based solely on cyprinodontiform fishes, rather than a broad survey of IGH across teleost fishes. While the current title and abstract as written will likely generate a broader readership, neither are accurate with respect to the true focus of the paper.

Constant regions should be precisely defined.

Referee: 3

Comments to the Author(s)

Dear authors,

Your manuscript presents well and describes a very interesting story. I have no major concerns regarding your study and my comments are minor.

Summarized they are:

Please address the issue of assembly errors in more detail in results/methods in addition to the part in the discussion. The IgH locus (loci) is tricky to handle for assembly algorithms and the V regions especially are vulnerable to collapse/breakage.

Please add detailed parameters for all bioinformatical approaches either in the main method or in the supplement.

Please consider your color scheme – it is somewhat strong on the eye and certain figures would be unreadable for colorblind persons.

Main text

Figure 2: this figure, although informative, is a bit hard to read. Color scheme is one element. The other are the expanded views. I would also like a detailed view of constant D and M on IgH2 in *N. furzeri*.

The reverse IgH2 region in *N. furzeri* could be interpreted as an assembly fault considering the data from the other closely related species and the sequence overview figure in the supplement. Could you add a sentence or two in the methods/results on how you proceeded to choose the most likely orientation of clones etc. in that region. This should also be reflected in the results/discussion as misassembly would affect some of the minor conclusions in your manuscript.

“Such expansion of IGHD through tandem duplications of the Cd2Cd3Cd4 exons is common in teleost loci fillatreau2013anstonishing” - Reference fillatreau2013anstonishing? Error in reference formatting?

Do you think that the number of V regions in *N. furzeri* is linked to the alternative assembly strategy? Have you by chance filtered out V regions in some way?

Consider toning down the conclusion on the three distinct lineages of IgHZ as there is a fork in your tree right at the base of two lineages: “suggests three distinct monophyletic clades of IgHZ”. Figure 6 may be not be the most intuitive to some. Maybe put it in supplement or have the alignment in the supplement so the reader can visualize?

When discussing the loss of IgHZ and mucosal immunity, what about IgD – if IgHs are so variable in their evolution, could not IgD maybe obtain a role related to mucosal linings? Also considering the genomic plasticity of fish immune systems gradually revealed through the last decade or so, could fish perhaps do without mucosal immunity as we know it? Recently, I saw a manuscript on BioRxiv by Mirete-Bachiller et al. presenting fish without IgH at all – how would they handle mucosal immunity? <https://doi.org/10.1101/793695>

Supplement:

Please add a “list of content” to the supplementary file for easier navigation.

I would like you to replicate figure S2 to for the IgH2 region as well as for IgM. I realize that the 100 % identity of some regions will make it difficult, but this region could be marked in the figure maybe?

Methods:

Please update accession where applicable - NFZ v2.0, Accession TBD

The majority of the bioinformatical methods described are lacking a description of parameters used etc. Either add to the manuscript or write the details in the supplement.

What is the underlying argument for retaining hits covering 1 % of the scaffold? In this case, you are eliminating all shorter scaffold. Your main focus is on constant regions and these tend to assemble well. However, you will likely lose some variable regions using this cutoff.

Please write out the full name of FLI in Jena.

A bit unclear: "In order to guarantee the reliability of the assembled scaffolds, the assemblies produced with BayesHammer- and QuorUM-corrected reads were compared, and scaffolds were broken into segments whose contiguity was agreed on between both assemblies". Please rephrase. I assume they were broken based on contiguous sections, but not quite sure.

When using the BAC clones to adjust your assemblies, did the BAC genomic structure take precedence over the genome assembly structure when verifying sequence within a scaffold? Or did you just use this strategy to combine scaffolds.

How many scaffolds were you not able to join in the end?

Author's Response to Decision Letter for (RSPB-2019-2180.R0)

See Appendix A.

RSPB-2020-0489.R0

Review form: Reviewer 2

Recommendation

Accept as is

Scientific importance: Is the manuscript an original and important contribution to its field?

Acceptable

General interest: Is the paper of sufficient general interest?

Marginal

Quality of the paper: Is the overall quality of the paper suitable?

Excellent

Is the length of the paper justified?

Yes

Should the paper be seen by a specialist statistical reviewer?

No

Do you have any concerns about statistical analyses in this paper? If so, please specify them explicitly in your report.

No

It is a condition of publication that authors make their supporting data, code and materials available - either as supplementary material or hosted in an external repository. Please rate, if applicable, the supporting data on the following criteria.

Is it accessible?

Yes

Is it clear?

Yes

Is it adequate?

Yes

Do you have any ethical concerns with this paper?

No

Comments to the Author

The authors have done an excellent job of responding to the previous reviews, particularly with respect to the addition of missing bioinformatic details. The current version of the manuscript is significantly improved. I have no further comments on the paper.

Decision letter (RSPB-2020-0489.R0)

09-Apr-2020

Dear Dr Valenzano

I am pleased to inform you that your manuscript RSPB-2020-0489 entitled "Extreme genomic volatility characterises the evolution of the immunoglobulin heavy chain locus in cyprinodontiform fishes" has been accepted for publication in Proceedings B.

The referee(s) have recommended publication, but also suggest some minor revisions to your manuscript. Therefore, I invite you to respond to the referee(s)' comments and revise your manuscript. Because the schedule for publication is very tight, it is a condition of publication that you submit the revised version of your manuscript within 7 days. If you do not think you will be able to meet this date please let us know.

Sincerely,

Dr Sarah Brosnan
Editor, Proceedings B
mailto: proceedingsb@royalsociety.org

Associate Editor
Board Member
Comments to Author:

Thank you for your care in responding to reviewer concerns. I have no additional suggestions for changes in the current document.

Reviewer(s)' Comments to Author:

Referee: 2

Comments to the Author(s).

The authors have done an excellent job of responding to the previous reviews, particularly with respect to the addition of missing bioinformatic details. The current version of the manuscript is significantly improved. I have no further comments on the paper.

Author's Response to Decision Letter for (RSPB-2020-0489.R0)

See Appendix B.

Decision letter (RSPB-2020-0489.R1)

14-Apr-2020

Dear Dr Valenzano

I am pleased to inform you that your manuscript entitled "Extreme genomic volatility characterises the evolution of the immunoglobulin heavy chain locus in cyprinodontiform fishes" has been accepted for publication in Proceedings B.

Open Access

Paper charges

Sincerely,

Proceedings B

Appendix A

MAX PLANCK INSTITUTE FOR **BIOLOGY OF AGEING**

MAX PLANCK INSTITUTE FOR **BIOLOGY OF AGEING** . Postfach 41 06 23 . D - 50866 Köln . Germany

Dario Riccardo Valenzano, PhD
Max Planck Research Group Leader
Evolutionary and Experimental Biology of Ageing
Max Planck Institute for Biology of Ageing
Joseph Stelzmann Str. 9b, 50931
Cologne, Germany
+49 (0)221 379 70 490
dvalenzano@age.mpg.de

March 3, 2020

1 / 12

Sarah Brosnan, PhD
Editor, *Proceedings of the Royal Society B*

Dear Dr. Brosnan,

Thank you very much for editing our manuscript entitled “*Extreme genomic volatility characterises the evolution of the immunoglobulin heavy chain locus in teleost fishes*“. We would also like to thank the three Referees for their very helpful suggestions and comments.

We have now generated an improved version of the manuscript, addressing all of the referees’ points, adding details on the bioinformatic analysis, and providing the new accession number for the genome. We provide an annotated manuscript (manuscript_annotated.pdf), which highlights all the changes in text and figures.

We believe that our revised manuscript addresses all the concerns and recommendations provided by the reviewers, and we thank you and the Reviewers for helping us improve the manuscript. We are greatly excited by this study, as we feel that by studying the evolution of the IgH locus in Cyprinodontiforms we can shed light on the evolution of adaptive immunity in vertebrates.

We hope that our revised manuscript is now acceptable for publication at *Proceedings of the Royal Society B*.

Thank you for editing our manuscript.
Please find enclosed below a point-by-point response to the reviewers’ comments.

Sincerely,

Dario Riccardo Valenzano, PhD

0. Editor comments

Your manuscript was reviewed by three experts and myself. While we all found the topic of interest, as you will see, there is some disagreement among the reviewers regarding the soundness of the analyses presented and their interpretation.

Two reviewers noted that the data were not accessible, including accession. This needs to be added should the paper be resubmitted.

Please pay careful attention to the concerns raised regarding the use of BLAST and the alternative splicing analysis. Again, multiple reviewers commented on the need for additional information on the bioinformatics approaches used, and one reviewer suggested that the authors could do a better job of placing the work in context of other studies on immune genes in fishes.

2 / 12

We thank the editor and the three referees for the enthusiasm regarding our work and for the constructive comments. We have now generated an improved version of the manuscript, addressing all of the referees' points, adding details on the bioinformatic analysis, and providing the new accession number for the genome. We hope now the manuscript is ready for publication on Proceedings of The Royal Society B.

1. Referee 1

1.1. This is a fascinating and well conducted study of the variation in genome content of IgM genes in several species of Cyprinodontiformes. The authors careful analysis shows variation in structure and expression of TM and secreted forms within single species and between clades.

We thank the reviewer for their supportive review of our work.

1.2 This not the only study of genome content variation in immune genes of fish – the authors should see the work of these authors on MHC and other immune gene families in zebrafish: McConnell and de Jong of Chicago, Jeff Yoder of North Carolina. It would be fascinating to see if there are parallels in the stories of gene loss and the effects on subsequent immune function in general.

We agree that it would be valuable to put this work in the broader context of teleost immune genes. We have added a paragraph on this to the end of the Discussion (lines 217-221), including key references from the authors the referee mentioned and others (Grimholt 2016, Dirscherl et al. 2014, Magadan et al. 2015).

1.3 On page 2 in the last paragraph, the authors say both species IgH loci are on chromosome 16. Isn't N. furzeri on chromosome 6?

We thank the referee for noticing this. We have corrected the error (line 67).

1.4 On page 3 in paragraph 2 there is a problem on line 13 “fillatreau2013astonishing” makes no sense.

Again, our thanks to the referee and we have corrected the error (line 79).

1.5 On page 5 and 7 the authors refer to the IgM locus as “primitive”, but it is clearly highly evolved, complex and flexible, so I am not sure it deserves that designation.

Our use of the term “primitive” here was intended in the phylogenetic sense of “inherited from the common ancestor” – i.e. present prior to the evolution of teleosts and shared with other vertebrate clades. This is in contrast to IgZ, which is a derived feature of teleosts and is absent elsewhere. Since the use of “primitive” in this sense is not important to the meaning of the sentences in question, we have removed it to avoid confusion (lines 148 and 195).

3 / 12

2. Referee 2

2.1 The authors present detailed information on the structure of the immunoglobulin heavy chain (IGH) locus in two cyprinodontiform fishes, and a less detailed analysis of the constant region in a broader suite of fishes of this family. The main focus area of the study is an important one for understanding adaptive immunity in vertebrates. While I applaud the authors on the work they have done to characterize IGH loci, I have several major concerns with the analyses and their interpretation outlined below.

We thank the reviewer for their constructive comments, and hope that our responses here and our revised manuscript convincingly address all the concerns raised.

2.2 The description of bioinformatic approaches for determining orthology is problematic and incomplete. Which BLAST algorithm was used? What were threshold values for initial searches?

We have now updated the supplementary information to include detailed supplementary methods (SI pp. 2-8), including detailed descriptions and parameters for all important commands, as well as software versions (Table S1, SI p. 21). We hope this additional information is satisfactory and are happy to provide further information as desired.

2.3 BLAST is notoriously bad for determining orthology, particularly when low complexity regions are involved. A much better approach would be using broader-scale syntenic information and proper orthology searches (e.g., OrthoMCL, Orthofinder, etc.). For example, a formal analysis of synteny with sequenced genomes would be very useful.

We used BLAST alignment algorithms at several points in this manuscript:

- First, we used nucleotide BLAST (BLASTN-Megablast) to identify scaffolds in the BAC insert assemblies arising from contaminating *E. coli* reads that were missed during the initial screen with Bowtie 2, by aligning assembled scaffolds to the *E. coli* genome (SI p. 2).
- Second, we used TBLASTN to obtain nucleic-acid sequences of stickleback and zebrafish constant-region exons, by aligning published amino-acid sequences to the nucleotide sequences of the published loci (SI p. 3).
- Third, we used BLASTP and BLASTN to align *IGH* gene sequences from reference species (zebrafish, stickleback and medaka) to the genomes of species for which we wanted to acquire new locus information, in order to identify scaffolds/inserts that might form part of the *IGH* locus sequence for each species. Similarly, we aligned reference gene sequences to assembled *N. furzeri* BAC inserts (SI p. 4)
- Fourth, we used nucleotide BLAST (again, BLASTN-MegaBLAST) to align candidate BAC inserts to candidate genome scaffolds during manual assembly of the complete *N. furzeri* *IGH* locus (SI p. 4).
- Fifth, and finally, we used BLASTN and BLASTP to reference constant-region exons to assembled *IGH* loci (in *N. furzeri* and *X. maculatus*) or candidate locus scaffolds (in other species) to identify the positions of CH exons and begin identifying intron/exon boundaries (SI p. 6).

All of these uses, including command-line specifications, are now described in our newly-added supplementary methods (SI pp. 2-8). It is not entirely clear to us which of these uses of BLAST the referee is commenting on in this point. While OrthoMCL and Orthofinder are both useful tools, they assume the availability of a set of protein sequences between which the researcher wants to identify orthology relationships. These tools might be an interesting alternative approach to e.g. the *Pachypanchax* CZ alignments in Fig. 6 (now Fig. S3, SI p. 9; c.f. point 3.10); however, in that figure we do not use BLAST, but rather perform exhaustive Needleman-Wunsch alignments between exon sequences. Most of our uses of BLAST in this manuscript are seeking to identify coding sequences in the first place, and as such would not be suitable for OrthoMCL or Orthofinder.

One important note is that the species tree displayed in Fig. 4 was not build based on assembled *IgH* loci. Rather, the tree displayed is a species tree based on previously-published phylogenies inferred from whole-genome information. We use it here to indicate relationships among the species for which we have provided novel *IgH* assemblies. As such, we are confident that synteny analysis of *IgH* loci will not substitute in this instance. We have edited the legend of Fig. 4 to make it clearer that the tree represents species relationships, and is not generated based on *IgH* locus assemblies.

2.4 Additionally, are there copy number variants / tandem duplications of *IGH* loci?

Tandem duplications of *IGH* loci are fairly common, with many species (e.g. stickleback and medaka) possessing multiple subloci in parallel (Bao et al. 2010, Gambón-Deza et al. 2010,

Magadán-Mompó et al. 2011). Antisense subloci are less common, but have previously been observed in teleost species, most notably in medaka. To our knowledge, most species possess only a single overall IGH locus on a single chromosome; however, salmonids like Atlantic salmon possess two paralogous IGH loci on different chromosomes (Yasuike et al. 2010).

In our case, our locus assembly was based primarily on a high-quality previously-published genome assembly (c.f. point 2.6) supplemented with high-coverage BAC sequencing and assembly (Supplementary Methods, SI pp. 2-4). Both the original source genome and the BAC assembly provide consistent evidence for the presence of two distinct subloci in antisense; see our response to point 3.6 for more information. While there is some evidence the presence of small numbers of isolated variable-region segments elsewhere in the genome (Table S2, SI p. 22), there is no evidence that these constitute intact *IgH* loci.

5 / 12

2.5 The alternative splicing analysis is problematic. Short Illumina fragments require assembly and are notoriously prone to errors, particularly in repetitive regions. A much better approach is to use long read, single molecule DNA sequencing technology (nanopore sequencing or minimally), such that sequence assembly can be simplified or altogether avoided.

We thank the reviewer for the critical feedback, and we understand the concern. We realise that we should have been clearer in explaining how this locus was assembled.

The *IgH* constant-region reference sequences used for the alternative splicing analyses are derived from previously-published genome sequences, both of which are based on multiple sources of data. In the case of *N. furzeri*, this included Illumina short reads, 10x genomics Gemcode 2 technology (which labels molecules longer than 100kb), as well as Nanopore sequencing. In the case of *X. maculatus*, both PacBio and high-coverage Illumina data were used (https://www.ncbi.nlm.nih.gov/assembly/GCF_002775205.1/).

Moreover, while the previously-published RNA-seq datasets used in these analyses were composed of Illumina reads, these were not assembled into a transcriptome but rather aligned directly onto the assembled constant-region sequences.

We have amended the manuscript to clarify the nature of the RNA-seq datasets used (line 289), and added supplementary methods describing the locus assembly and RNA-seq alignment pipelines in more detail (SI pp. 6-7)

2.6 The genomes used for the analysis are not available and thus it is not possible to review the methods used here. How complete are these assemblies? What technology was used for sequencing? How complete were the gene models (e.g., BUSCO percentage?).

At the time of original submission, the publication featuring the improved *N. furzeri* genome assembly was still in preparation, and as such the data were not yet publicly available. The publication is now available on bioRxiv (<https://doi.org/10.1101/852368>); the genome has been submitted to GenBank and will be available at the BioProject accession PRJNA599375, which

we have added to the manuscript (lines 68 and 226, plus Table S4, SI p. 22). A detailed description of the quality of that assembly is given in that paper; the gene content assessment using BUSCO method gives 95.20% complete BUSCOs. Moreover, mapping Genbank *N. furzeri* RefSeq RNA to the new assembly to predict gene models gives a 96.1% for complete BUSCOs.

2.7 The title is misleading. The data are based solely on cyprinodontiform fishes, rather than a broad survey of IGH across teleost fishes. While the current title and abstract as written will likely generate a broader readership, neither are accurate with respect to the true focus of the paper.

We understand the point regarding the article title, and have now replaced “teleost fishes” with “cyprinodontiform fishes” in consequence. However, we feel the abstract as written does accurately reflect the contents of the manuscript.

2.8 Constant regions should be precisely defined.

Unfortunately, the precise meaning of this remark is not entirely clear to us, preventing us from providing a meaningful answer.

3. Referee 3

3.1 Your manuscript presents well and describes a very interesting story. I have no major concerns regarding your study and my comments are minor.

We thank the reviewer for their supportive review of our work.

3.2 Please address the issue of assembly errors in more detail in results/methods in addition to the part in the discussion. The IgH locus (loci) is tricky to handle for assembly algorithms and the V regions especially are vulnerable to collapse/breakage.

With the exception of the BAC inserts used in the construction of the *N. furzeri* locus, all genome sequences used in this study were obtained from high-quality, previously published genome assemblies. The BACs, meanwhile, were sequenced at very high coverage (Table S3, SI p. 22) and underwent extensive manual validation prior to integration into the assembly. In addition, for most species, we focused exclusively on the constant regions, which are less repetitive, and hence less prone to misassembly, than the variable regions; we therefore largely feel comfortable relying on the published assemblies in these cases.

The main regions of potential concern with respect to this issue are therefore the variable regions of the *N. furzeri* and *X. maculatus*. In the case of *N. furzeri*, many of the V segments, especially in *IGH1*, are validated through presence in multiple sequences (e.g. one or more BACs plus the genome sequence – see new Table S7, SI p. 25); however, this validation is not available for *X. maculatus* and we are relying entirely on the quality of the pre-existing genome

assembly, which was published by a different group. It is possible that this assembly is missing some V segments. We have now included a cautionary note regarding the *X. maculatus* locus in the results (line 102).

One way to check the completeness of our assemblies would be to perform immunoglobulin sequencing on these species and use V-haplotype-reconstruction tools like IgDiscover to see whether any new V-segments can be identified. External validation of this kind would certainly be very valuable, and we have added a brief note to that effect in the discussion (line 210); however, this work is outside the scope of the present study.

3.3 Please add detailed parameters for all bioinformatical approaches either in the main method or in the supplement.

7 / 12

We have now updated the supplementary information to include detailed supplementary methods (SI pp. 2-8), including detailed descriptions and parameters for all important commands, as well as software versions (Table S1, SI p. 21). We hope this additional information is satisfactory and are happy to provide further information as desired.

3.4 Please consider your color scheme – it is somewhat strong on the eye and certain figures would be unreadable for colorblind persons.

We have adjusted the colour scheme of Fig. 2, S1 and S2 – specifically the colours of the V, D and J segments – to improve readability for colourblind persons.

*3.5 Figure 2: this figure, although informative, is a bit hard to read. Color scheme is one element. The other are the expanded views. I would also like a detailed view of constant D and M on IgH2 in *N. furzeri*.*

As requested, we have added an additional supplementary figure showing the expanded view of *N. furzeri* IGH2 (Fig. S1, SI p. 9).

*3.6 The reverse IgH2 region in *N. furzeri* could be interpreted as an assembly fault considering the data from the other closely related species and the sequence overview figure in the supplement. Could you add a sentence or two in the methods/results on how you proceeded to choose the most likely orientation of clones etc. in that region. This should also be reflected in the results/discussion as misassembly would affect some of the minor conclusions in your manuscript.*

This is an important question. What evidence do we have for the presence of two subloci in opposite sense?

- The underlying genome assembly, which is high quality (c.f. points 2.5 and 2.6), contains the entirety of the IGH1 region in forward orientation, followed by a gap,

followed by a single V segment (IGH2V1-01) in antisense. This provides an initial hint that there is a second sublocus present in antisense.

- One of the key BAC inserts in the assembly (276N03) aligns to the region of the genome assembly downstream of IGH2V1-01, extends upstream into a gap in the genome assembly, and contains more complete V-segments in antisense in this gap-covering region.
- Another key BAC insert (209K12) contains parts of both subloci, present in opposite sense on a single insert. As discussed above, this insert was assembled at very high coverage (>1000x, Table S3, SI p. 22) and validated through manual end-to-end PCR and Sanger sequencing of assembly fragments, and is highly robust. This therefore constitutes strong evidence of two subloci in opposite sense.
- Finally, other species also have antisense subloci in their IGH loci. In particular, as shown in Figure 4, *Nothobranchius orthonotus* (a close sister species of *N. furzeri*) also contains IGHD constant-region exons on a single scaffold (scf33917), strongly suggesting a similar configuration to the *N. furzeri* locus and lending credence to our assembly of the latter. An antisense sublocus is also seen in medaka (both in the original characterisation from Magadán-Mompó et al. 2011 and our new characterisation presented here), and inverted regions are also found in the Atlantic salmon IGH loci.

8 / 12

Many of these points are now reflected in figure S7 (SI p. 13), which we have modified to indicate the positions of the two subloci within the locus assembly. We hope this helps clarify this important issue.

3.7 “Such expansion of IGHD through tandem duplications of the Cd2Cd3Cd4 exons is common in teleost loci fillatreau2013anstonishing” - Reference fillatreau2013anstonishing? Error in reference formatting?

We thank the reviewer for pointing this out. We apologise for the oversight and have corrected the error (line 79).

3.8 Do you think that the number of V regions in N. furzeri is linked to the alternative assembly strategy? Have you by chance filtered out V regions in some way?

For the *N. furzeri* locus, the VH segments in *IGH1* are mostly present in both the underlying genome assembly and the BAC inserts overlapping with that assembly (e.g. 277J10). The only exceptions to this are IGH1V1-06 and IGH1V4-02p (which are present in multiple BACs but not the genome assembly; see point 3.19) and IGH1V1-07. For IGH2, this is not the case: except for IGH2V1-01 (which is also present on the genome assembly), all VH segments are only present on BAC 276N03. This lack of independent validation presents a higher opportunity for error; however, as discussed in the response to point 3.6, the BACs were sequenced at very high coverage and underwent extensive manual checking during the process of assembly. It is, of course, still possible that some V regions are missing; as discussed in point 3.6, one promising way to check this would be using immunoglobulin-sequencing data and V-haplotype

inference tools like IgDiscover, and we have added a brief note to that effect in the discussion (line 210); however, we believe this is beyond the scope of the current paper.

In addition, it is worth noting that, while the total number of functional VH segments in the *N. furzeri* locus is unusually small for a teleost species, the number of VH segments per sublocus is more in line with previously-published assemblies from related species. As assembled, the *N. furzeri* locus contains 17 VH segments in IGH1 and 7 in IGH2; in comparison, the previously published medaka locus contains 2 to 12 VH segments per sublocus, while the previously published stickleback locus contains 6 to 18. The small total in *N. furzeri* therefore stems more from a smaller number of subloci than a paucity of VH segments per sublocus. The question of course remains of why the number of subloci in *N. furzeri* is smaller than in these species; while we excluded discussion of this from the paper due to limited space, we believe this is an interesting question worthy of further consideration in the future.

3.9 Consider toning down the conclusion on the three distinct lineages of IgHZ as there is a fork in your tree right at the base of two lineages: “suggests three distinct monophyletic clades of IgHz”.

We thank the reviewer for this suggestion and have made appropriate changes at several points in the manuscript (lines 50, 142 and 148).

3.10 Figure 6 may be not be the most intuitive to some. Maybe put it in supplement or have the alignment in the supplement so the reader can visualize?

As suggested, we have now moved Figure 6 to the supplementary information (now Fig. S3, SI p. 9).

3.11 When discussing the loss of IgHZ and mucosal immunity, what about IgD – if IgHs are so variable in their evolution, could not IgD maybe obtain a role related to mucosal linings? Also considering the genomic plasticity of fish immune systems gradually revealed through the last decade or so, could fish perhaps do without mucosal immunity as we know it? Recently, I saw a manuscript on BioRxiv by Mirete-Bachiller et al. presenting fish without IgH at all – how would they handle mucosal immunity? <https://doi.org/10.1101/793695>

We thank the referee for these stimulating comments! We agree that there are many possibilities for how IGHZ-lacking species might handle mucosal immunity, including through IGHD or even relying on systems other than the adaptive immune system. We have added brief comments to the relevant part of the discussion reflecting these considerations (lines 195-197).

3.12 Please add a “list of content” to the supplementary file for easier navigation.

We have now added a table of contents at the start of the supplementary information (SI p. 1).

3.13 I would like you to replicate figure S2 to for the IgH2 region as well as for IgM. I realize that the 100 % identity of some regions will make it difficult, but this region could be marked in the figure maybe?

As requested, we have repeated the RNA-seq splicing analysis for *N. furzeri* IGH2; the results can be viewed in Figure S6 (SI p. 12).

3.14 Please update accession where applicable - NFZ v2.0, Accession TBD

The genome has been submitted to GenBank and will be made available under the BioProject ID PRJNA599375. The genome assembly and annotation are still being processed by GenBank; we will make the GenBank genome accession itself available as soon as possible; in the meantime we have added the BioProject accession to the manuscript (lines 68 and 226, plus Table S4, SI p. 22).

10 / 12

3.15 The majority of the bioinformatical methods described are lacking a description of parameters used etc. Either add to the manuscript or write the details in the supplement.

As discussed in point 3.3, we have now added detailed supplementary methods, including detailed parameters, to the supplementary information (SI pp. 2-8).

3.16 What is the underlying argument for retaining hits covering 1 % of the scaffold? In this case, you are eliminating all shorter scaffold. Your main focus is on constant regions and these tend to assemble well. However, you will likely lose some variable regions using this cutoff.

During the initial scaffold-selection step, any genome scaffold that did not contain any alignments to IGH gene segments was rejected, while any scaffold aligning to more than one class of gene segments (e.g. V and J, J and C, or two distinct C exon types) was retained; the proximity of multiple segment types was considered a strong signal of a locus candidate.

The 1% cutoff the referee refers to applied only to the remaining scaffolds, i.e. those aligning to exactly one type of IGH gene segments. These scaffolds were only retained if these alignments covered at least 1% of the total length of the scaffold; this weak cutoff was intended to eliminate long scaffolds and superscaffolds containing a small number of orphaned gene segments, as in this case the total proportion of the scaffold covered by the alignments would be small. Conversely, a small number of gene-segment alignments was often sufficient to cover >1% of short scaffolds, which were consequently retained for downstream analysis.

We thank the reviewer for raising this concern. We have updated the manuscript text to reflect the explanation above (lines 224-229), and we hope this new version is clearer.

3.17 Please write out the full name of FLI in Jena.

Done (lines 232 and 252).

3.18 A bit unclear: “In order to guarantee the reliability of the assembled scaffolds, the assemblies produced with BayesHammer- and QuorUM-corrected reads were compared, and scaffolds were broken into segments whose contiguity was agreed on between both assemblies”. Please rephrase. I assume they were broken based on contiguous sections, but not quite sure.

We believe the referee is correct in their understanding of our methods here; to avoid confusion, we have rephrased the relevant part of the manuscript as follows (lines 238-240):

“The two scaffolded assemblies (using BayesHammer- and QuorUM-corrected reads, respectively) agreed on the order and orientation of contigs at many points, but disagreed at others. In order to guarantee the reliability of the assembled scaffolds, the assemblies were broken into segments, such that the contiguity of each segment was agreed on between both assemblies.”

11 / 12

We hope this new phrasing is clearer.

3.19 When using the BAC clones to adjust your assemblies, did the BAC genomic structure take precedence over the genome assembly structure when verifying sequence within a scaffold? Or did you just use this strategy to combine scaffolds.

In most cases, the genome sequence took precedence over BAC sequences in the event of conflicts. However, in response to concerns about segment loss similar to those the reviewer expresses above, we took the position that false-negative exclusion of segments was particularly undesirable during locus assembly, and hence allowed the BAC sequences to take precedence when they contained a segment that was missing from the genomic sequence. In the event, this only occurred once: a region containing two VH segments, IGH1V1-06 and IGH1V4-02p, was present in multiple BACs but not on the genome assembly, and we included this region in the locus assembly to avoid losing these two segments. In all other cases, either the genome sequence took precedence or BACs were used to fill in gaps in the genome assembly (including the joining of scaffolds).

We have added an additional table to the supplementary information containing information on which VH segments are present on which sequences contributing to the locus sequence (Table S7, SI p. 25), and have added information from the final sentence above to the supplementary methods (SI, p. 4, paragraph 8).

3.20 How many scaffolds were you not able to join in the end?

In total, out of the six genome scaffolds and eleven BAC inserts identified as potentially containing part of the locus sequence, a total of four scaffolds and six BACs were discarded. We have added two supplementary tables (S2 and S3, SI p. 22) giving details of which BACs and scaffolds were identified as candidates and which were eventually included. In the case of the

genome scaffolds, none of the discarded sequences contained any putative constant-region sequences or more than 3 putative variable-region gene segments.

MAX PLANCK INSTITUTE FOR **BIOLOGY OF AGEING** · Postfach 41 06 23 · D - 50866 Köln · Germany

Dario Riccardo Valenzano, PhD
Max Planck Research Group Leader
Evolutionary and Experimental Biology of Ageing
Max Planck Institute for Biology of Ageing
Joseph Stelzmann Str. 9b, 50931
Cologne, Germany
+49 (0)221 379 70 490
dvalenzano@age.mpg.de

April 14, 2020

Sarah Brosnan, PhD
Editor, *Proceedings of the Royal Society B*

1 / 1

Dear Dr. Brosnan,

Thank you very much again for editing our manuscript entitled “*Extreme genomic volatility characterises the evolution of the immunoglobulin heavy chain locus in cyprinodontiform fishes*“. We would also like to also thank again the three Referees for their very helpful suggestions and comments.

We are glad that our manuscript RSPB-2020-0489 has been accepted for publication in *Proceedings of the Royal Society of London B*.

We have now uploaded all the files as requested.
Here I am providing a short non-technical summary of our work:

Vertebrates have evolved a lymphocyte-based adaptive immune system, which deploys precise and long-lasting immune responses against a vast range of antigens. Teleost fishes possess sophisticated adaptive immune systems; however, the complex gene structures underlying adaptive immunity are only known for a limited number of teleost species, and little is known about their evolution. To expand our understanding of the evolution of adaptive immunity, we reconstructed the immunoglobulin heavy chain (*IGH*) gene loci of several related cyprinodontiform fishes, discovering multiple independent duplications and deletions of the specialised antibody class *IGHZ* and revealing extreme volatility in the evolution of *IGH* locus structure.

Thank you again for editing our manuscript.

Sincerely,

Dario Riccardo Valenzano, PhD